# Retrieval and Timing Performance of Chewing-Based Eating Event Detection in Wearable Sensors

**DOI:** 10.3390/s20020557

**Published:** 2020-01-20

**Authors:** Rui Zhang, Oliver Amft

**Affiliations:** Chair of Digital Health, Friedrich-Alexander Universität Erlangen-Nürnberg (FAU), Henkestraße 91, 91052 Erlangen, Germany; oliver.amft@fau.de

**Keywords:** automated dietary monitoring, eating detection, eating timing error analysis, biomedical signal processing, smart eyeglasses, wearable health monitoring

## Abstract

We present an eating detection algorithm for wearable sensors based on first detecting chewing cycles and subsequently estimating eating phases. We term the corresponding algorithm class as a bottom-up approach. We evaluated the algorithm using electromyographic (EMG) recordings from diet-monitoring eyeglasses in free-living and compared the bottom-up approach against two top-down algorithms. We show that the F1 score was no longer the primary relevant evaluation metric when retrieval rates exceeded approx. 90%. Instead, detection timing errors provided more important insight into detection performance. In 122 hours of free-living EMG data from 10 participants, a total of 44 eating occasions were detected, with a maximum F1 score of 99.2%. Average detection timing errors of the bottom-up algorithm were 2.4 ± 0.4 s and 4.3 ± 0.4 s for the start and end of eating occasions, respectively. Our bottom-up algorithm has the potential to work with different wearable sensors that provide chewing cycle data. We suggest that the research community report timing errors (e.g., using the metrics described in this work).

## 1. Introduction

Eating occasion detection is at the core of automated dietary monitoring (ADM) in humans, targeting healthy diet management [1,2]. We regard intake to consume food pieces with dietary activities including ingestion, chewing, and swallowing [3] as an eating occasion if all dietary activities start and end in a given temporal relation. Meals or snacks are typical examples of eating occasions. Eating occasions thus have a start and end denoting the timing of intake beginning and intake completion. For solid and semi-solid food, chewing (i.e., the cyclic opening and closing of the jaw) is typically the longest activity within eating occasions [3]. We therefore consider chewing as representative of eating occasions, denoted as *eating events* in this work.

Recording chewing to interpret eating has been attempted in a variety of approaches intended for free-living ADM (see Section 2), as accurate eating event timing detection is essential for diet management. For example, users could be reminded to check vital parameters such as glucose level when the initial moment of an eating event is detected. Similarly, users could be asked to confirm food details or take a photo of leftovers immediately after an eating event ends. In both examples it is important that timing errors of the eating event detection are minimal. Hence, timing errors determine whether an eating event detection approach is suitable across the ADM application spectrum.

Detecting dietary activities, including eating events, in wearable or ambient sensor data is a complex pattern analysis and modelling problem due to the inter- and intra-individual variability in free-living behaviour patterns. Approaches to eating event detection and analysis can be categorised as top-down or bottom-up sensor data processing: In the top-down approach, eating events are detected by applying sliding windows to the sensor time series and applying feature pattern models. If necessary, further information details such as chewing cycles, intake gestures, etc. could be derived using the detected eating events. Conversely, in a bottom-up approach, individual dietary activities are modelled and the result is subsequently used to detect eating events. The early abstraction in bottom-up processing may help to deal with varying dietary activity patterns. Furthermore, bottom-up processing fits into hierarchical data processing schemes of resource-constrained wearable and IoT systems, where instead of raw data, derived parameters or events are communicated between system components.

This investigation proposes a bottom-up eating detection algorithm and compares it with two top-down algorithms. The bottom-up eating detection algorithm first detects individual chewing cycles. Retrieved chewing cycles are then used to detect eating events and estimate start and end of eating occasions. In contrast, top-down algorithms apply sliding windows over the sensor time series to detect eating events. The bottom-up algorithm proposed here is potentially agnostic to the particular sensor used, as long as chewing cycle information is acquired. In particular, the following contributions are made:We present a bottom-up algorithm for eating event detection based on chewing time-series data. The algorithm works based on chewing cycle information and has only four parameters.We evaluate and compare bottom-up and top-down eating event detection algorithms in data of a free-living study, where participants continuously wore unobtrusive diet monitoring eyeglasses. The diet eyeglasses recorded electromyographic (EMG) data of the temporalis muscles. We analysed retrieval performance as well as start and end timing errors of detected eating events.We describe and analyse a procedure to derive eating event reference data in a free-living context. Our approach combines participant self-reports with a mostly unobtrusive chewing reference measurement. The analysis confirms that our reference estimation approach reached a timing resolution of less than one second in free-living behaviour data.

## 2. Related Work

ADM has received increasing research interest over the last decade, where eating event detection based on data from various body-worn and ambient sensors has been frequently considered. Most investigations that considered quantitative performance for eating event detection focused on detection accuracy or retrieval metrics. In this investigation, we highlight that timing errors are critical for detection performance and investigate timing errors specifically.

Eating event detection has often been approached by top-down data processing. For example, Dong et al. used a wrist motion sensor to detect eating, reporting 81% accuracy in 449 hours of free-living data [4]. Thomaz et al. also used a wrist-worn three-axis accelerometer to monitor eating in free-living conditions [5]. The random forest classifier yielded 66% precision and 88% recall for one day of data and intra-individual analysis. Bi et al. implemented a headband carrying a bone-conducting acoustic sensor and reported eating detection performance of over 90% [6]. Farooq et al. used accelerometer-equipped eyeglasses to detect food intake in the lab and in short-term free-living [7]. The highest F1 score of 87.9% ± 13.8% (mean ± standard deviation) was achieved with a 20 s sliding window using a *k*-nearest neighbour classifier. Studies involving multiple sensor modalities are a recent trend in eating event detection applications. Wahl et al. implemented an eyeglasses prototype equipped with an inertial measurement unit (IMU), an ambient light sensor, and a photoplethysmogram (PPG)sensor for the recognition of nine daily activities, including eating [8]. The classification reached an average accuracy of 77%. Merck et al. realised a multi-device monitoring system involving in-ear audio, head motion, and wrist motion sensors, which could recognise eating with 92% precision and 89% recall [9]. Papapanagiotou et al. proposed an ear-worn eating monitoring system based on PPG, audio and accelerometer, achieving an accuracy up to 93.8% and class-weighted accuracy up to 89.2% in eating detection [10]. Bedri et al. used an ear-worn system for chewing instance detection. An F1 score of over 80% and accuracy of over 93% was reported [11]. Timing error for eating start was 65.4 s. The authors did not report the timing error at eating ends. Doulah et al. investigated the effect of the temporalresolution of eating microstructure analysis, including the duration of eating events [12]. The analysis did not yield insight into start and end time estimates for eating events. In our prior investigation of top-down eating detection based on free-living EMG recordings, a one-class support vector machine (ocSVM) yielded an F1 score of 95%. Timing error analysis showed 21.8 ± 29.9 s for eating start and 14.7 ± 7.1 s for eating end [13].

For the bottom-up data processing approach, dietary activities that characterise eating are modelled, and eating is subsequently derived from these activities. Chewing has frequently been investigated as a basis for subsequent eating analysis. Amft et al. investigated chewing detection for ADM using an ear-plug acoustic sensor, capturing vibration patterns during chewing [14]. Bedri et al. proposed earwear using proximity sensors for the detection of tiny deformations of the outer ear during chewing [15]. Eating could be detected with 95.3% accuracy with a user-dependent classification. Zhang et al. was the first to use smart eyeglasses to detect chewing, analysing EMG electrode positions in eyeglasses frames and the effect of hair on the EMG signal [16]. EMG electrodes were embedded into the eyeglasses’ temples, and chewing cycles were detected with a precision and recall of 80%. In subsequent work [17], a refined version of the eyeglasses was used for eating detection, yielding an accuracy of above 95% in natural, free-living data. Furthermore, it was demonstrated that soft foods such as banana provide identifiable EMG signatures. Chung et al. incorporated a force-sensitive load cell in eyeglasses hinges to monitor temple movement during chewing, head movement, talking, and winking. A classification of these activities yielded an F1 score of 94% [18]. Farooq et al. attached a strain sensor at the temporalis muscle area to obtain chewing cycle information [19]. With additional accelerometer data, the authors reported an F1 score of 99.85% for recognising eating from other physical activities in laboratory recordings.

So far, timing performance has been rarely reported, partly because methods to derive eating reference in free-living studies were missing. Here, we evaluated three algorithms in free-living EMG recordings with a realistic ratio of eating vs. non-eating time. All algorithms can be used with one or more sensors and in multimodal configurations. In particular, the bottom-up algorithm builds on chewing cycle information extracted from sensor data, and thus can be applied with other sensors besides EMG by adapting the chewing cycle extraction. Our current work focuses in particular on the analysis of timing errors.

## 3. Eating Event Detection Algorithms

We propose a bottom-up eating event detection algorithm and compare it to two top-down algorithms. As input for all algorithms we consider a multi-source sensor data stream of chewing cycle measurements, corresponding to a random process $X_{n}\left(t\right)$, where *n* indexes the random variables (e.g., sensor channels or features) and *t* is the time index. For example, the sensor could be an EMG monitor measuring the temporalis muscle contraction or acoustic transducers measuring vibration patterns due to food fracture. An overview of the algorithm pipelines for all algorithms considered is shown in Figure 1. Below, we formally describe the algorithms.

### 3.1. Bottom-Up Algorithm

The idea of this algorithm is to estimate eating events from the density of chewing cycles, where a relatively high frequency of chewing cycles indicates eating. After pre-processing multi-source sensor signals $X_{n}\left(t\right)$, chewing cycle onsets $C_{n}$ were detected. Subsequently, a sliding window of length $w_{0}$, was applied around each retrieved onset of $C_{n}$ (i.e., with a step size of one onset). Then, the sliding window moved to the next detected onset. At every onset, we calculated chewing cycle frequency $f_{n}$ as the number of detected onsets per time interval $w_{0}$. A chewing segment start $t_{n,\;\mathrm{start}}$ was detected as the first onset in $C_{n}$ at the signal start or an onset after a preceding detected chewing segment, where $f_{n}$ equalled or exceeded $\theta_{0}$. The end of a chewing segment $t_{n,\;\mathrm{end}}$ was determined as the onset in $C_{n}$ where $f_{n}$ equalled $\theta_{0}$ and the $\left(\theta_{0}-1\right)$-th subsequent $f_{n}$ equalled 1. Detection results of *n* sensor sources were combined and post-processed by eliminating gaps between adjacent groups of chewing segments. The details of each step are described below.

#### 3.1.1. Signal Pre-Processing

Pre-processing steps vary depending on the type of sensors used. It is likely that the human body acts as an antenna and picks up power line noise. Thus, we applied a notch filter to raw signal $X_{n}\left(t\right)$ to eliminate potential power line interference at frequency $f_{\mathrm{nf}}$. In this study, we used dual-channel smart eyeglasses EMG data sampled at 256 Hz per channel. Hence, $X_{n}\left(t\right)\left(n=1,2\right)$ represents EMG data in this case. The notch filter frequency was set to $f_{\mathrm{nf}}=$ 50 Hz. Baseline wander and motion artifacts were removed using a high-pass filter with a cut-off frequency of $f_{\mathrm{hpf}}=$ 20 Hz—a typical value for EMG signal processing. The resulting data $X_{n,\;\mathrm{hpf}}$ were rectified for detection. The pre-processed and rectified data were abbreviated as $X_{n}$. The pseudo code is in Algorithm block 1.
**Algorithm block 1**: Signal pre-processing.**Input:** 
Multi-source free-living data Xn(t)**Parameter:** Notch filter band-stop frequency fnf, high-pass filter cut-off frequency fhpf**Output:** Pre-processed data Xn1:Xn,nf=NotchFilt(Xn(t),fnf)2:Xn,hpf=HighPassFilt(Xn,nf,fhpf)3:Xn=|Xn,hpf|


#### 3.1.2. Chewing Cycle Detection

Chewing cycle detection was performed by adapting the EMG onset detection principle initially proposed by Abbink et al. [20]. Every chewing cycle has an onset time corresponding to the moment when the muscle contraction starts, and an offset time corresponding to the contraction end. Hence, the number of onsets should represent the number of chewing cycles. First, a sliding window of size *w* was applied to Xn. The value of *w* should be no larger than the duration of a typical chewing cycle. Here we used 0.4 s (100 samples for the EMG signal), as chewing cycle frequency typically ranges between 0.94 and 2.17 Hz [21]. We derived a conditional summation of sensor samples within the window: For samples 0 to *w*/2 within the current window starting at i0, we derived index1=∑i=0w/21ifXn[i0+i]<θC. For samples in the second half-window, index2=∑i=w/2+1w1ifXn[i0+i]>θC was summed. Finally, index=index1+index2 was derived. Parameter θC was set to μ+3×σ, where μ was the mean and σ the standard deviation derived from baseline noise of Xn. Both μ and σ were estimated across training data of all participants. The amplitude of the baseline noise was assumed to be Gaussian distributed and threshold θC was set to cover 99% of the confidence interval. As the window with size *w* was slid with a step size of one sample, an index in range [0,w] was obtained for each sample, forming a new time series In per signal source *n*. To determine chewing onsets, we derived points of In that exceeded θP×w, with θP in the range [0,1]. Considering the chewing frequency, the temporal distance between neighbouring detection points of In should be larger than tinterval=1/3 s. Detected chewing cycle onsets were sequentially saved in a list Cn. The pseudo code is shown in Algorithm block 2.
**Algorithm block 2**: Chewing cycle detection.**Input:** Pre-processed data Xn**Parameter:** EMG burst threshold θC, sliding window size *w*, peak threshold θP, peak interval tinterval**Output:** 
A list of detected chewing cycle onsets Cn1:index=0,In←∅,Cn←∅2:**for**(i=1,i<w/2,i++)**do**3:    **if**
Xn[i]<θC
**then**4:        index+=15:**for**(i=w/2,i<w,i++)**do**6:    **if**
$X_{n}\left[i\right]>\theta_{C}$
**then**7:        index+=18:**for**$\left(i=w/2+1,\;i<\mathrm{length}\left(X_{n}\right)-w/2,\;i++\right)$**do**9:    **if**
$X_{n}\left[i-w/2-1\right]<\theta_{C}$
**then**10:        index−=111:    **if**
$X_{n}\left[i-1\right]<\theta_{C}$
**then**12:        index+=113:    **if**
$X_{n}\left[i-1\right]>\theta_{C}$
**then**14:        index−=115:    **if**
$X_{n}\left[i+w/2\right]>\theta_{C}$
**then**16:        index+=117:    $I_{n}$.append(index)18:**for**$\left(i=0,\;i<length\left(I_{n}\right)-2,\;i++\right)$**do**19:    **if**
$I_{i}<I_{i}+1$ and $I_{i}+1>I_{i}+2$ and $I_{i}+1>\theta_{P}$
**then**20:        $C_{n}.append\left(i+1\right)$21:        $i+=t_{\mathrm{interval}}$


#### 3.1.3. Chewing Segment Detection

We applied a sliding window of size w0 to Cn, with the start of the window located at the first chewing cycle onset Cn[0], and subsequently slid to the adjacent onset until reaching the end of Cn. With the window starting at Cn[j], the chewing cycles in the window were counted and noted as the *j*th chewing cycle frequency fn[j]. We applied a criterion fn[j]≥θ0 to confirm that onset Cn[j] belonged to a chewing segment. Correspondingly, the first onset in Cn that also satisfied the criterion fn[jstart]≥θ0 was considered as the start of the first chewing segment tn,start[0]. An onset with fn[jend]=θ0 and fn[jend+θ0−1]=1 indicated that Cn[jend+θ0−1] was the only onset in the latest window, that is, the final onset/end of the *k*-th estimated chewing segment, denoted as tn,end[k]. The next onset after tn,end[k] that satisfied the criterion fn[j]≥θ0 was considered as the (k+1)-th chewing segment start tn,start[k+1]. The pseudo code is shown in Algorithm block 3.
**Algorithm block 3**: Chewing segment detection.**Input:** List of detected chewing cycle onsets Cn**Parameter:** Sliding window size w0, chewing cycle frequency threshold θ0**Output:** Detected chewing segment starts and ends (tn,start, tn,end) from each signal source *n*1:tn,start←∅,tn,end←∅2:**function**Find_Start_and_End(Cn,j,θ0,w0)3:    fn,end = onset count in interval [Cn[j], Cn[j]+w0]4:    **if**
fn,end==1
**then**5:        tn,end.append(Cn[j+θ0−1])6:        **for**
(i=j+θ0,i<length(Cn),i++)
**do**7:           fn,start = onset count in interval [Cn[i], Cn[i]+w0]8:           **if**
fn,start>=θ0
**then**9:               tn,start.append(Cn[i])10:               break11:        **return** i12:    **else**13:        Find_Start_and_End(Cn,j0+fn,end+θ0−1,θ0,w0)14:**for**(j=1,j<length(Cn),j++)**do**15:    fn[j]=onsetcountininterval[Cn[j],Cn[j]+w0]16:    fn[j+1]=onsetcountininterval[Cn[j+1],Cn[j+1]+w0]17:    **if**
tn,start==∅ and fn[j]>=θ0
**then**18:        tn,start.append(Cn[j])19:    **if**
fn[j]>=θ0 and fn[j+1]<θ0
**then**20:        step = FindStartEnd(Cn,j,θ0,w0)21:        j+=step+θ0−1

#### 3.1.4. Fusion of Multi-Source Detection

The fusion of *N* sensor or feature channels was made by taking the union of source-specific chewing segments:(1)$$T_{\mathrm{merge}}=\overset{N}{\underset{n=1}{⋃}}\overset{K_{n}}{\underset{k=1}{⋃}}\left[t_{n,\;\mathrm{start}}\left[k\right],\;t_{n,\;\mathrm{end}}\left[k\right]\right],$$
where $T_{\mathrm{merge}}$ was a list of the merged chewing segments of *N* sources, and $K_{n}$ was the number of chewing segments in Channel *n*. All detected segments were collected chronologically regardless of any overlapping among sources. For the evaluation data used in this investigation, bilateral EMG channels yielded two lists of chewing segments. Hence, N=2.

#### 3.1.5. Gap Elimination

In free-living, eating is often accompanied by interrupts (e.g., conversations). Thus, an eating event is usually represented by several chewing segments in $T_{\mathrm{merge}}$, where the gaps indicate interrupts without chewing cycles. Depending on the detection application and choice of eating event definition, it is reasonable to combine temporally close segments into one final eating event. We denote the start and end of the *k*-th segment $T_{\mathrm{seg}}\left[k\right]$ in $T_{\mathrm{merge}}$ as start[k] and end[k] respectively, and the gap between $T_{\mathrm{seg}}\left[k\right]$ and $T_{\mathrm{seg}}\left[k+1\right]$ as $T_{\mathrm{gap}}\left[k\right]$. We generated a new list $T_{\mathrm{concatenated}}$ by removing all gaps that were smaller than $t_{\mathrm{gap}}$:(2)$$T_{\mathrm{concatenated}}=\underset{k\in S}{⋃}\left(T_{\mathrm{seg}}\left[k\right]\cup T_{\mathrm{gap}}\left[k\right]\cup T_{\mathrm{seg}}\left[k+1\right]\right),$$
where
(3)$$S=\{k|start\left[k+1\right]-end\left[k\right]<t_{\mathrm{gap}}\}.$$
An estimated eating event start ${\hat{T}}_{\mathrm{start}}\left[q\right]$ and end ${\hat{T}}_{\mathrm{end}}\left[q\right]$ with (q=1,2,…,Q) were thus obtained as the start and end of every segment in $T_{\mathrm{concatenated}}$, where *Q* was the number of segments (i.e., detected eating events) in $T_{\mathrm{concatenated}}$. In the present investigation, $t_{\mathrm{gap}}$ was set to 5 min.

### 3.2. Top-Down Algorithms

Two top-down algorithm variants were considered with different chewing segment detection blocks (see Figure 1): Threshold-based top-down and ocSVM top-down. Several blocks of the top-down and bottom-up pipelines were identical, including signal pre-processing (Section 3.1.1), fusion of multi-source detection (Section 3.1.4), and gap elimination (Section 3.1.5). Here we concentrate on the individual variants of the chewing segment detection.

#### 3.2.1. Threshold-Based Top-Down Algorithm

A sliding window of size w1 and step size s1 was applied to Xn. We computed the chewing intensity feature *F* in each sliding window and applied threshold θ1. If F>θ1, the window was reported as chewing. For the present investigation, we considered EMG readings as time series containing chewing information and extracted EMG work as chewing intensity feature *F*. EMG work was defined as the summation of rectified EMG samples within the sliding window. For the EMG data, s1 was 256 samples (1 s). The pseudo code is shown in Algorithm block 4.
**Algorithm block 4**: Chewing segment detection.**Input:** Preprocessed signals Xn**Parameter:** Sliding window size w1, window step size s1, chewing intensity feature threshold θ1**Output:** Detected eating starts/ends from each signal source *n*: tn,start and tn,end1:tn,start←∅, tn,end←∅2:**for**(i=s1,i<length(Xn)−w1,i+=s1)**do**3:    extractFpreviousfromXn[i−s1:i+w1−s1]4:    extractFcurrentfromXn[i:i+w1]5:    extractFnextfromXn[i+s1:i+w1+s1]6:    **if**
Fprevious<θ1 and Fcurrent>θ1
**then**7:        tn,start.append(*i*)8:    **if**
Fcurrent>θ1 and Fnext<θ1
**then**9:        tn,end.append(i+s1)


#### 3.2.2. ocSVM Top-Down Algorithm

We applied a non-overlapping sliding window of size $w_{2}$ to the EMG data. An ocSVM model was trained based on the windows to detect chewing segments using the same features as described in [13]. The radial basis function (RBF) was used as the kernel. The hyper-parameters γ and ν were varied, where γ weighted the non-support vectors’ influence on the hyper plane, and ν was an upper bound on the fraction of margin errors as well as a lower bound of the fraction of support vectors relative to the number of training samples. The ocSVM predicted the class of each sliding window as either eating or non-eating.

## 4. Evaluation Methodology

We evaluated the algorithms using a free-living dataset collected from smart eyeglasses with integrated EMG electrodes. Details of the eyeglasses design and data collection process can be found in [17]. Here we summarise the relevant data collection procedures, as well as evaluation methods.

### 4.1. Participants and Recording Protocol

The dataset was collected from a group of 10 participants (6 male, 4 female, average age of 25.1 years, average BMI of 23.8 kg/m${}^{2}$) each wearing the smart eyeglasses for one day of regular activity without script or specific protocol. The study was approved by the Ethical Committee of FAU Erlangen-Nürnberg. All participants were healthy and consented to participate after having received oral and written study information.

Each participant received a pair of 3D-printed smart eyeglasses mechanically fitted to their head using a personalisation procedure similar to [22], ensuring that the effect of hair, loss of contact between skin and electrodes, or movement was minimal. In each temple of the eyeglasses frame, dry stainless-steel electrodes of 3 mm × 20 mm (EL-DRY-STEEL-5-20, BITalino, Lisbon, Portugal) were integrated, yielding a two-channel EMG recording system on each side of the head. The EMG electrode pairs were positioned to capture activity of the temporalis muscle. A reference EMG channel was recorded from the right temporalis muscle via gel electrodes attached to the skin at the corresponding forehead region. All EMG channels were acquired with an EMG recorder (ACTIWAVE, CamNtech, Cambridgeshire, United Kingdom) at a sampling rate of 256 Hz per channel.

Participants were suggested to wear the eyeglasses during one entire recording day (i.e., attaching the system right after getting up and ending before going to bed at night). Recordings were conducted in free-living conditions without dietary constraints. Participants chose their diets and conducted other daily activities at their choice. Participants were asked to log activities in a paper-based 24-h activity journal with 1 min resolution, including any food intake as well as start and end times of eating events. As Figure 2 show:

### 4.2. Data Corpus

By the end of the recording, we collected a total of 122.3 h of free-living data including 44 eating events ranging from 54 s to 35.8 min, which summed up to 429 min of eating for all participants combined. Eating took up 5.8% of the whole dataset. Participants took off eyeglasses for a total time of 12 min during the recordings, which corresponds to 0.16% of the total recordings. Known activities reported by participants in the activity journal included cooking, eating, walking, transportation, attending lectures, performing office work, having conversations, doing housework, brushing teeth, playing video games, going to the cinema, and engaging in physical exercise. Through visual inspection we observed various artefacts in the data corpus including, for example, suspected teeth grinding [17].

### 4.3. Free-Living Eating/Non-Eating Reference Construction

Obtaining accurate reference information on eating events in unsupervised free-living studies is particularly challenging. Here, we propose a combination of participant activity journal and EMG reference recordings. All eating events were annotated using a custom Matlab annotation software. Our annotation process comprised two steps: coarse manual annotation using the activity journal and fine-tuning through reference EMG recordings. Coarse manual annotation was realised by searching the journal for the participant-logged start time $T_{\mathrm{start}}\left[i\right]$ and end time $T_{\mathrm{end}}\left[i\right]$ of each annotated eating event, indexed *i*. As manual journaling is often imprecise in identifying event times, a fine-tuning step was used to adjust coarse eating event times: Start and end times $T_{S}\left[i\right]$ and $T_{E}\left[i\right]$ of eating event *i* were adjusted by visually searching the reference EMG data for chewing cycle patterns in the neighbourhood of approx. ± 1 min (journal resolution) around the coarse annotations $T_{\mathrm{start}}\left[i\right]$ and $T_{\mathrm{end}}\left[i\right]$. Since each chewing cycle had a duration of around 1/3 s, the fine-tuned eating event labels $T_{S}\left[i\right]$ and $T_{E}\left[i\right]$ resulted in a chew-accurate eating/non-eating reference with resolution of approximately 1/3 s. The derived start and end times were considered as eating/non-eating reference for algorithm evaluation. The eating/non-eating reference construction is illustrated in Figure 3.

Type 1 errors (false positives) could occur in the eating/non-eating reference if an activity journal entry could not be matched to any chewing-like pattern in the reference EMG signal. We inspected all entries in the participant journal and compared them to the reference EMG signal. In the present dataset, all participant-annotated events could be matched to the EMG reference.

Type 2 errors (false negatives) could occur in the eating/non-eating reference if participants omitted annotations. To amend potential omissions from the activity journal, we first inspected the entire reference EMG data for chewing-like signal patterns that did not correspond to any entry in the journal. For each chewing-like pattern found, we inspected the activity journal to obtain insight into the participant’s momentary context. We observed that concise activations in the EMG reference occurred occasionally without corresponding eating annotations (e.g., during a lecture). Yet, EMG activations were typically short (i.e., less than five consecutive activations with lower EMG work compared to confirmed chewing). Given a non-eating context and the clear non-chewing signal patterns, we attributed the activations to teeth grinding. Jaw motion during speaking does not involve profound temporalis muscle activation, as there is hardly any teeth clenching and thus substantially lower EMG work than during chewing [16]. In addition, non-chewing muscle activity is typically non-periodic, thus observable and distinguishable during time series inspection. Overall, we did not find Type 2 errors in the dataset, supporting our eating/non-eating reference construction approach for free-living recordings.

### 4.4. Evaluation Metrics

A grid search over the window length parameters $w_{i}$ and thresholds $\theta_{i}$ with i=0,1,2, and $\theta_{2}=\left(\gamma ,\nu \right)$ representing the combination of the ocSVM hyper-parameters was performed to investigate optimal parameter combinations. To evaluate the eating event detection algorithms, we derived the overlap between retrieved eating events and any eating/non-eating reference label. The precision and recall of each algorithm were calculated according to: $Recall=\frac{T_{\mathrm{tp}}}{T_{\mathrm{gt}}}$ and $Precision=\frac{T_{\mathrm{tp}}}{T_{\mathrm{ret}}}$, where $T_{\mathrm{gt}}$ was the summed duration of all *P* eating events according to the constructed eating/non-eating reference labels, calculated as:(4)$$T_{\mathrm{gt}}=\sum_{p=1}^{P}\left(T_{\mathrm{end}}\left[p\right]-T_{\mathrm{start}}\left[p\right]\right),$$
while $T_{\mathrm{ret}}$ was the summed duration of all *Q* detected eating events by the algorithm:(5)$$T_{\mathrm{ret}}=\sum_{q=1}^{Q}\left({\hat{T}}_{\mathrm{end}}\left[q\right]-{\hat{T}}_{\mathrm{start}}\left[q\right]\right),$$
and $T_{\mathrm{tp}}$ was the summed overlap duration between retrieved eating events and the eating/non-eating reference:(6)$$T_{\mathrm{tp}}=\sum_{p=1}^{P}\sum_{q=1}^{Q}\left(\min\left(T_{\mathrm{end}}\left[p\right],{\hat{T}}_{\mathrm{end}}\left[q\right]\right)-\max\left(T_{\mathrm{start}}\left[p\right],{\hat{T}}_{\mathrm{start}}\left[q\right]\right)\right),$$
given the following premise:(7)$$\min\left(T_{\mathrm{end}}\left[p\right],{\hat{T}}_{\mathrm{end}}\left[q\right]\right)-\max\left(T_{\mathrm{start}}\left[p\right],{\hat{T}}_{\mathrm{start}}\left[q\right]\right)>0.$$
${\hat{T}}_{\mathrm{end}}\left[q\right]$ and ${\hat{T}}_{\mathrm{start}}\left[q\right]$ were the start and end time points of the *q*th retrieved eating event, *Q* was the number of retrieved eating events, and *P* was the number of eating events in the eating/non-eating reference. All times were computed at a resolution of 1 sample (1/256 s). Finally, the F1 score was calculated as the harmonic mean of precision and recall.

The evaluation was performed using leave-one-participant-out (LOPO) cross-validation. In each evaluation fold, the EMG data were split into a training set of nine participants and a test set of one participant. This process was repeated 10 times until every participant’s data were in the test set once. Training data were used in a grid search to estimate performance under different parameter combinations. Optimal parameter combinations were chosen according to the training data performance and applied with the test data to estimate algorithm performance. The test results of all folds were averaged to obtain the total algorithm performance. For the bottom-up algorithm, $w_{0}$, $\theta_{0}$, and $\theta_{P}$ were analysed. For the threshold-based top-down algorithm, $w_{1}$ and $\theta_{1}$ were analysed, and for the ocSVM top-down algorithm, $w_{2}$, γ, and ν were analysed.

### 4.5. Detection Timing Errors

We further investigated the detection timing error of every algorithm. The average start and end timing errors of the algorithms were calculated as follows:(8)$$\Delta {\bar{T}}_{S}=\frac{\sum_{q=1}^{Q}\min\left(\left.|{\hat{T}}_{S}\left[q\right]-T_{S}\left[p\right]\right|{|}_{p=1,2,\ldots ,P}\right)}{Q},$$
and
(9)$$\Delta {\bar{T}}_{E}=\frac{\sum_{q=1}^{Q}\min\left(\left.|{\hat{T}}_{E}\left[q\right]-T_{E}\left[p\right]\right|{|}_{p=1,2,\ldots ,P}\right)}{Q}.$$
$\Delta {\bar{T}}_{S}$ and $\Delta {\bar{T}}_{E}$ were the average absolute detection errors at the start and end of eating events.

To investigate retrieval performance in detail and identify the algorithms’ behaviour, different optimisation objectives were analysed. Using the grid search over the parameter space, the best performance point according to maximal F1 score (termed $P_{X}$), minimal start timing error $\Delta {\bar{T}}_{S}$ (termed $P_{S}$), and minimal end timing error $\Delta {\bar{T}}_{E}$ (termed $P_{E}$) were derived.

## 5. Results

Algorithm detection performances according to the test data are shown in Figure 4 for varying parameter combinations. The threshold-based top-down algorithm could not reach meaningful F1 scores, indicating that detecting eating events is not a trivial task. The performance map of the ocSVM algorithm shows a periodic landscape due to the variation of parameters γ and ν. The best performance of the bottom-up algorithm was achieved with $\theta_{P}=0.7$. The bottom-up algorithm had a smooth landscape across the parameters. For all algorithms, the three performance points ($P_{X}$, $P_{S}$, $P_{E}$), did not coincide at the same parameter settings. To illustrate the performance points quantitatively, they are summarised in Table 1. The bottom-up algorithm yielded comparable performance values across all performance points ($P_{X}$, $P_{S}$, $P_{E}$). At best, the bottom-up algorithm reached an F1 score of 99.2%, yielding a start/end error ($\Delta {\bar{T}}_{S}$ and $\Delta {\bar{T}}_{E}$) of 2.4±0.4 s and 4.3±0.4 s, respectively. The results show that the bottom-up algorithm outperformed the top-down algorithms.

Figure 5 shows the effect of varying the peak detection threshold $\theta_{P}$ of the bottom-up algorithm, indicating robust retrieval and timing performance ($P_{X}$, $P_{S}$, $P_{E}$) for a parameter range of $0.65<\theta_{P}<0.8$. The best retrieval and timing performances were achieved at $\theta_{P}=0.7$.

Figure 6 illustrates retrieved eating events as point pairs across F1 scores, where the line ends represent average start and end timing errors ($\Delta {\bar{T}}_{S}$ and $\Delta {\bar{T}}_{E}$). $\Delta {\bar{T}}_{S}$ and $\Delta {\bar{T}}_{E}$ were obtained by varying the algorithm parameters and averaging the individual timing errors obtained for specific retrieval performances. For the bottom-up algorithm, the graph shows the performance obtained by varying sliding window size $w_{0}$ and chewing cycle frequency threshold $\theta_{0}$ at fixed peak detection threshold $\theta_{P}=0.7$. There was no parameter combination for the threshold-based top-down algorithm that yielded an F1 score above 40%. In contrast, bottom-up and ocSVM top-down algorithms provided retrieval performances of up to 99% and 95% respectively. With increasing F1 score, timing errors tended to decline. It can be derived from Figure 6 that the relation between start and end timing errors varied between algorithms. For the bottom-up algorithm and F1 score >80%, the start timing error $\Delta {\bar{T}}_{S}$ became smaller than the end timing error $\Delta {\bar{T}}_{E}$.

Figure 7 shows examples of the detected eating event starts and ends. The bottom-up algorithm yielded similar detected labels to the eating/non-eating reference whereas the ocSVM top-down algorithm incurred larger timing errors for some eating event instances.

## 6. Discussion

The F1 score describes the algorithm’s retrieval performance by retrieved and missed eating instances, while timing errors reveal the accuracy of estimated event timing. Considering the varying eating durations in a free-living context, the two metrics are not necessarily similar in their sensitivity, thus we argue here that both are relevant metrics for evaluation. Among the few investigations on event timing in ADM, Dong et al. [4] reported event start-timing errors of 0.6 minutes, and end errors of 1.5 min. The authors determined intake from bites using arm motion, while the present investigation was based on chewing. Bedri et al. [11] evaluated eating event detection using a metric called delay, measuring the time from the beginning of an eating event until it was recognised. The average delay reported was 65.4 s. In contrast to the investigation of Bedri et al. [11], we also evaluated the timing error at the end of eating events. Our bottom-up algorithm yielded average start/end timing errors of 2.4 s and 4.3 s.

We believe that the bottom-up method is practically useful for eating event start and end detection, as well as, for example, sending reminders, sampling user responses, and gathering environmental variables. Study participants did not complain or reject wearing the eyeglasses for one day. Hence, the combination of the bottom-up algorithm and smart eyeglasses could be adopted in unconstrained free-living applications. In contrast to several previous investigations of eating detection that require the training of many parameters, our bottom-up approach requires that only four parameters be set ($w_{0}$, $\theta_{0}$, $\theta_{P}$, and $t_{\mathrm{gap}}$). Our analysis indicates that performance was unaffected by parameter changes across a wide value range (i.e., shown as a smooth performance space in Figure 4). Pattern learning may work reliably when trained on sufficient data with proper features. Considering the variability in free-living behaviour and the unbalanced distribution of eating and non-eating times, substantial training data is needed to implement any learning method and therefore a minimal number of free parameters is key. The bottom-up method outperformed our top-down methods, with a higher F1 score and lower detection timing errors. We attribute the higher performance yielded in the present investigation to the expert knowledge incorporated in the bottom-up approach.

In both top-down and bottom-up methods, the sliding window size $w_{i}$ influenced the algorithm performance. In top-down methods, a small sliding window of length $w_{i}$ contained fewer data samples, which usually led to less representative features. Thus, the lowest timing errors were typically not achieved with smallest sliding window sizes (e.g., $w_{i}<10\;s$). Similarly, in the bottom-up method, both window size $w_{0}$ and the second parameter $\theta_{0}$ influenced the detection performance. Hence, a small window size $w_{0}$ did not always give the best performance.

The timing errors of top-down methods were highly dependent on the combination of sliding window length and window step size. Large sliding window sizes included more dietary activity information, but usually failed in accurately detecting the starts and ends of eating events as the window was filled with both eating and non-eating data. Figure 7 shows impressively that the ocSVM top-down algorithm indeed incurred larger timing errors due to the larger sliding window size. In our previous investigation [13], we adopted window overlaps and majority voting on windows with differing results. We observed that retrieval performances differed marginally when comparing overlapping and non-overlapping windowing approaches. The bottom-up algorithm was not affected by the window parameterisation problem, as the window step size is determined by distance of neighbouring chewing onsets. Thus, eating and non-eating rarely coincided in one window.

The bottom-up algorithm is based on chewing cycle detection, which decouples the eating event detection from the sensor type. The detection leverages event frequency information (i.e., chewing cycle frequencies), which can be obtained with different chewing monitoring approaches. We expect that the algorithms could be applied with various sensors or sources that provide chewing cycle information, including acoustics [1], ear canal deformation [15], strain on head skin [19], eyeglasses temple motion [18], etc.

The present investigation analysed relevant free parameters of the proposed algorithms to determine their stability. For example, the sweep of the peak detection threshold $\theta_{P}$ showed desirable performance trends (Figure 5) allowing us to set $\theta_{P}$ to a proper range—approximately [0.65,0.8]. In addition, the pipeline block “gap elimination” used the parameter $t_{\mathrm{gap}}=5$ min to merge temporally close eating detections. The parameter $t_{\mathrm{gap}}$ supports our informal definition of eating events as temporally linked sequences of dietary activities during one meal or snack [3] and was set based on experience. Varying $t_{\mathrm{gap}}$ means to change the representation of eating occasions (i.e., meals and snacks), which is outside of the scope of this investigation.

While this investigation focuses on the retrieval performance, the computational complexity of the algorithms is an important consideration for wearable resource-limited systems. In a detection, the computational complexity is O(n) for the threshold-based top-down algorithm, and $O(n_{\mathrm{sv}}\times n)$ for the ocSVM top-down algorithm. Here, *n* is the input data dimension and $n_{\mathrm{sv}}$ is the number of support vectors of the ocSVM model. The complexity of the bottom-up algorithm is decided by the chewing cycle detection method. For the proposed bottom-up algorithm, the corresponding complexity is O(n). With a proper chewing cycle detection approach, the bottom-up algorithm is suitable to execute, for example, on wearables at a minimal computational cost. The delay due to processing was not addressed in this investigation. However, with the low complexity of all algorithms, processing delay is expected to have a negligible effect compared to the algorithm timing errors.

This investigation was supported by a new method to obtain reference data on eating times in a free-living context, where we combined the participants’ activity journals with reference EMG measurements. While the activity journals yielded rather coarse timing, they provided us with context information on the users’ behaviour. The reference EMG measurement complemented the journal with accurate timing resolution of individual chewing cycles. However, adherence to journals is known to decline quickly over several days of measurement [23]. Hence, it is reasonable to assume that journals alone would be too inaccurate. We avoided video recordings to retrieve eating/non-eating reference due to privacy concerns and the potential impact of cameras on natural, free-living behaviour.

One limitation of our study is that only young healthy participants were involved. For other populations, the eating structure could vary, which could generate different eating durations. However, our present investigation already showed that eating events ranging from short snacks of 54 s to 35.8 min meals could be recognised. Other populations may benefit from different pre-processing steps or other sensors to apply the discussed bottom-up algorithm. We are planning longer-term studies in the future.

## 7. Conclusions

We proposed a bottom-up eating event detection algorithm that uses chewing cycle information as input and compared it to two top-down algorithms, including threshold-based and ocSVM algorithms. Evaluation of the algorithms was performed using free-living data with smart eyeglasses recording EMG data bilaterally from the temporalis muscles. Our results indicate that the F1 score became less meaningful at high retrieval rates above 0.9. The analysis of timing errors revealed substantial differences of several tens to hundreds of seconds on average between top-down and bottom-up algorithms. The grid search analysis showed smooth performance transitions during parameter variation for the bottom-up algorithm. We conclude that timing error analysis is an important component in performance estimation, besides a relevant retrieval metric, as the F1 score. We suggest that the research community report timing errors (e.g., using the metrics described in this work). The bottom-up algorithm yielded the overall best results with the lowest timing errors of 2.4±0.4 s for eating start and 4.3±0.4 s for eating end. The bottom-up algorithm is thus suitable for eating event detection.

## Figures and Tables

**Figure 1 sensors-20-00557-f001:**
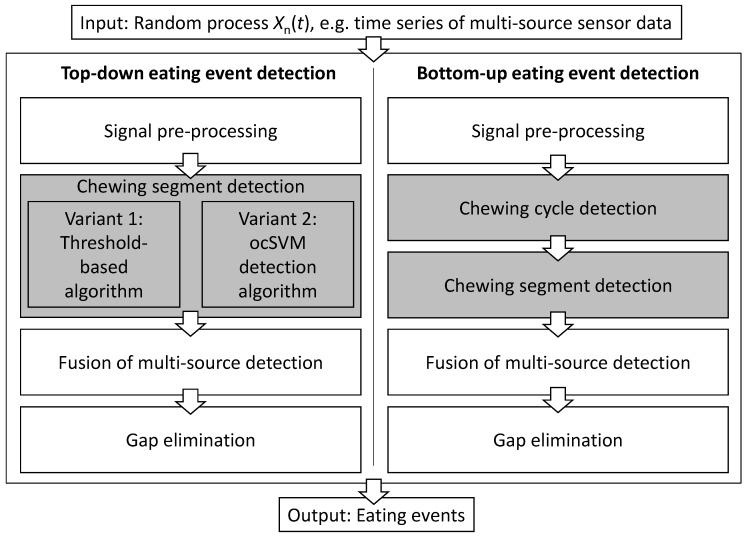
Overview of the top-down and bottom-up eating event detection algorithms investigated in this work. White processing blocks indicate functions shared by the algorithms. Shaded processing blocks are specific functions for each algorithm. Both top-down algorithms follow the same detection pipeline with different implementations of the “Chewing segment detection” block. ocSVM: one-class support vector machine.

**Figure 2 sensors-20-00557-f002:**
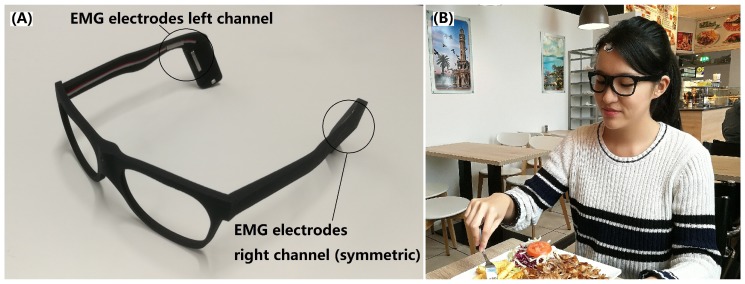
Illustration of the EMG eyeglasses and study: (**A**) Eyeglasses frame with electromyographic (EMG) electrodes symmetrically integrated on the temples. (**B**) Study participant wearing the EMG eyeglasses. Reference EMG electrodes were attached to the skin at the right forehead temporalis muscle position.

**Figure 3 sensors-20-00557-f003:**
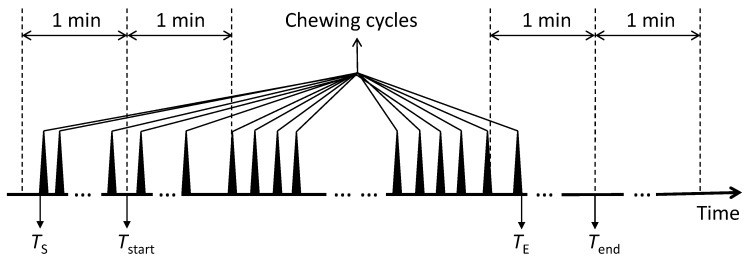
Illustration of the free-living eating/non-eating reference construction. $T_{\mathrm{start}}$ and $T_{\mathrm{end}}$ are start and end times of an eating event obtained from the participant journal, while $T_{S}$ and $T_{E}$ are the corrected start and end times derived by searching the EMG reference ± 1 min around $T_{\mathrm{start}}$ and $T_{\mathrm{end}}$. The eating/non-eating reference construction is described in Section 4.3.

**Figure 4 sensors-20-00557-f004:**
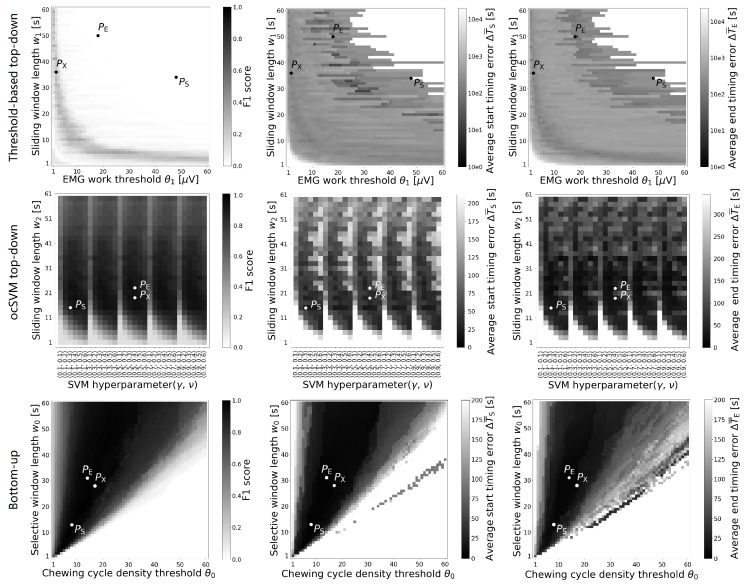
F1 score, average start and end timing errors for test data and each eating event detection algorithm using grid search over the parameter space. The highest F1 score location was denoted as $P_{X}$, while $P_{S}$ and $P_{E}$ indicate the minimal start timing error $\Delta {\bar{T}}_{S}$ and minimal end timing error $\Delta {\bar{T}}_{E}$, respectively. The bottom-up algorithm performance was obtained with fixed peak detection threshold $\theta_{P}=0.7$.

**Figure 5 sensors-20-00557-f005:**
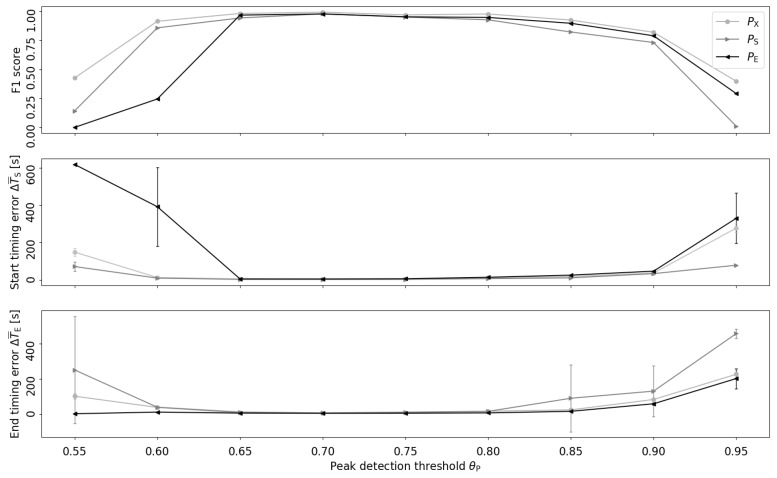
Retrieval and timing performance of the bottom-up algorithm at different peak detection thresholds $\theta_{P}$. In the timing error diagrams, caps on vertical line ends indicate the standard deviation.

**Figure 6 sensors-20-00557-f006:**
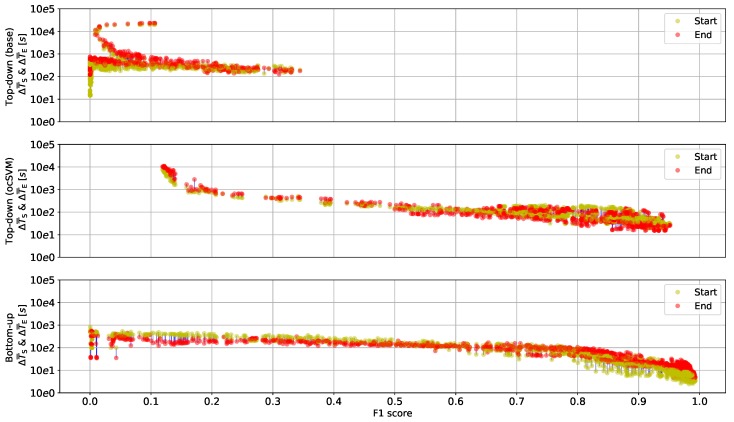
Relation of retrieval and timing performance of all three algorithms. $\Delta {\bar{T}}_{S}$ and $\Delta {\bar{T}}_{E}$ were obtained by varying algorithm parameters. Blue lines link average start and end timing errors of all eating events at a given algorithm parameter set. With increasing F1 score, timing errors declined. Note that timing error analysis could be performed only for eating events retrieved by an algorithm. The bottom-up algorithm ($\theta_{P}=0.7$) achieved the highest F1 score at smallest timing errors among all algorithms investigated. Point pairs were down-sampled for visualisation.

**Figure 7 sensors-20-00557-f007:**
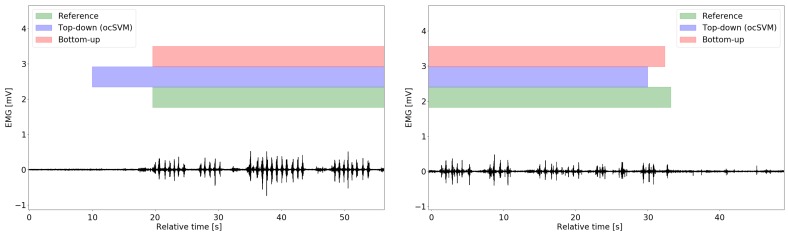
Examples of data situations with the corresponding retrieval results of bottom-up and ocSVM top-down algorithms obtained at each algorithm’s performance point $P_{S}$ (left column) and $P_{E}$ (right column). As the diagrams illustrate, the ocSVM algorithm may anticipate or delay eating events’ starts, as ocSVM deploys a time-domain sliding windowing with a given step size, whereas the bottom-up algorithm did not.

**Table 1 sensors-20-00557-t001:** Performance comparison among algorithms using optimal parameter settings for each performance point ($P_{X}$, $P_{S}$, $P_{E}$). For timing metrics, mean performance ± std. dev. are shown. For example, the bottom-up algorithm reached an F1 score of 99.2% at best, where the start/end error was 2.4±0.4 s and 4.3±0.4 s, respectively.

Metric		Performance Points		
		*P* _X_	*P* _S_	*P* _E_
F1 score (%)	Threshold-based top-down	36.7	0.03	0.001
	ocSVM top-down	95.1	90.9	93.2
	Bottom-up	99.2	97.8	97.7
$\Delta {\bar{T}}_{E}(s)$	Threshold-based top-down	152.4 ± 21.7	10.1 ± 3.0	185.9 ± 35.9
	ocSVM top-down	30.0 ± 36.4	18.8 ± 27.9	53.2 ± 61.7
	Bottom-up	3.0 ± 0.6	2.4 ± 0.4	4.8 ± 2.9
$\Delta {\bar{T}}_{E}(s)$	Threshold-based top-down	177.4 ± 12.1	265.8 ± 86.5	63.0 ± 11.9
	ocSVM top-down	25.9 ± 39.4	26.9 ± 38.3	15.2 ± 19.0
	Bottom-up	4.9 ± 0.3	6.4 ± 0.5	4.3 ± 0.4

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
