# Peer review of "Retrieval and Timing Performance of Chewing-Based Eating Event Detection in Wearable Sensors"

_sensors, 2020, doi:10.3390/s20020557_

Round 1

Reviewer 1 Report

This work presented an eating detection algorithm.  The authors evaluated the algorithm using EMC recording from the Temporalis muscles.  With this algorithm, it shows a maximum F1 score of 99.2% from 10 participants, with 44 eating occasions in total. 

The authors provided an excellent introduction, which includes the background knowledge of eating occasion detection and current methods.  Limitation for each method has been briefly discussed.  The authors have also explained clearly where the proposed method come from and what issues the method trying to solve.  The organization and sections of this manuscript are consistent with the proposed contributions.  The figures and writings are both in high quality.  Overall, it is a high standard paper and the topic is a good fit to Sensors. 

Some suggestions are listed below:

It would be good to clarify the difference and improvement of this research from the previous publications of [16] and [17]. It may be easier for readers to understand, if a photo is provided to show how the eyeglass looks like and how the sensor data was collected from participants. At the end of discussion, the authors mentioned about one limitation of the study that only young healthy participants were involved. Besides that, is there any other limitation or any parts that could be improved in the future?

Reviewer 2 Report

In this paper the authors have proposed a bottom-up eating event detection algorithm that uses chewing cycle information as input and compared it to two top-down algorithms.

This is an interesting paper, well structured and organized. The proposed approach is novel in various regards and is supported by data analysis.

The authors may wish to briefly discuss if their approach would be easily adopted in real life scenarios and if there would be any issues with invasiveness in data gathering phase.

Reviewer 3 Report

The authors developed a faster processing algorithm for eating event detection that is promising for wearable healthcare devices and human-machine interface using a lower computational cost. In this study, wearable 3D-printed eyeglasses were created for EMG data acquisition from the movements of the Temporalis muscles.  Although the processing time and detection accuracy are good, the algorithm needs several empirically chosen parameters and the data sets are still small. In other words, it is yet suitable for real-time or dynamic detection in wearable applications. The authors showed up limitations of their algorithm (mainly due to the test participants’ privacy) and also the possibility of further improvement using a combination of multiple sensor types. Consequently, this work can be considered for publication in Sensors journal.

The authors should show images of the EMG electrodes integrated to the 3D-printed eyeglasses. What is the EMG electrode size? If the EMG electrodes and sensors are commercial products, please mention in the main text. Is there any effect of hair or eyeglasses movement (head-shaking, glasses adjustment, etc.) on the data recording of the EMG electrodes?

The authors stated in page 10/18: “Jaw motion during speaking does not involve profound Temporalis muscle activation as there is hardly any teeth clenching and thus substantially lower EMG work than during chewing.” The authors should present the raw data or any reference to confirm this statement. Besides, in the situation that the test participants drink or eat soft food, can the EMG electrodes record the Temporalis muscle movements?

In line 143 – page 5/18: “First a sliding window of size w was applied to Xn. The value of w should be no larger than the duration of a typical chewing cycle, here we used 0.4 s as chewing activity mainly occurred in the range of 0.94 to 2.17 Hz.” How much was the actual interval time between 2 data points of the developed wearable measurement system (shown in Fig 6)? If that interval time is about 0.1 s or more (as in Fig 3, each pixel as a data point), the window size w = 0.4 s may not contain enough data points to evaluate some parameters (as stated in line 333 – page 15/18: “Pattern learning may work reliably when trained on sufficient data with proper features.”)

(a) From page 4/18 to page 11/18, some parameters were not well defined and a few ones had multiple symbols. What is the difference among θ0, θ1, θ, and θP? (b) Were the mean (µ) and the standard deviation (σ) of the Xn baseline noise, which were used to set θ, obtained by fitting and calculation? (c) What is the difference between Kn and Q (section 3.14-3.15 and 4.4)?

As stated in line 365 - page 16/18: “The parameter tgap supports our informal definition … was set based on experience. Varying tgap … is out of scope for this investigation.”, the statement in line 329 – page 15/18 may be not reliable (“In contrast to several previous investigations of eating detection that require training of many parameters, our bottom-up approach does require fitting three parameters (w0, θ0, and θP) only.)
